# Activation of Nrf2 in Mice Causes Early Microvascular Cyclooxygenase-Dependent Oxidative Stress and Enhanced Contractility

**DOI:** 10.3390/antiox11050845

**Published:** 2022-04-26

**Authors:** Dan Wang, Cheng Wang, Xueqin Hao, Gabriela Carter, Rafaela Carter, William J. Welch, Christopher S. Wilcox

**Affiliations:** 1Division of Nephrology and Hypertension and Hypertension Center, Georgetown University, Washington, DC 20007, USA; dw32@georgetown.edu (D.W.); gcarte17@alumni.jh.edu (G.C.); rcarter2021@visi.org (R.C.); welchw@georgetown.edu (W.J.W.); 2Division of Nephrology, Department of Medicine, The Fifth Affiliated Hospital of Sun Yat-sen University, Zhuhai 519000, China; wangch2@mail.sysu.edu.cn; 3Department of Human Anatomy and Histoembryology, School of Basic Medical Sciences, Henan University of Science and Technology, Luoyang 471023, China; haoxueqin@haust.edu.cn

**Keywords:** tert-butylhydroxyquinone (tBHQ), thromboxane, bardoxolone methyl, hypertension, endothelin

## Abstract

Nuclear factor erythroid factor E2-related factor 2 (Nrf2) transcribes antioxidant genes that reduce the blood pressure (BP), yet its activation with tert-butylhydroquinone (tBHQ) in mice infused with angiotensin II (Ang II) increased mean arterial pressure (MAP) over the first 4 days of the infusion. Since tBHQ enhanced cyclooxygenase (COX) 2 expression in vascular smooth muscle cells (VSMCs), we tested the hypothesis that tBHQ administration during an ongoing Ang II infusion causes an early increase in microvascular COX-dependent reactive oxygen species (ROS) and contractility. Mesenteric microarteriolar contractility was assessed on a myograph, and ROS by RatioMaster™. Three days of oral tBHQ administration during the infusion of Ang II increased the mesenteric microarteriolar mRNA for p47^phox^, the endothelin type A receptor and thromboxane A_2_ synthase, and increased the excretion of 8-isoprostane F_2α_ and the microarteriolar ROS and contractions to a thromboxane A_2_ (TxA_2_) agonist (U-46,619) and endothelin 1 (ET1). These were all prevented in Nrf2 knockout mice. Moreover, the increases in ROS and contractility were prevented in COX1 knockout mice with blockade of COX2 and by blockade of thromboxane prostanoid receptors (TPRs). In conclusion, the activation of Nrf2 over 3 days of Ang II infusion enhances microarteriolar ROS and contractility, which are dependent on COX1, COX2 and TPRs. Therefore, the blockade of these pathways may diminish the early adverse cardiovascular disease events that have been recorded during the initiation of Nrf2 therapy.

## 1. Introduction

Nuclear factor erythroid factor E2-related factor 2 (Nrf2) binds to its chaperone Kelch-like ECH-associated (Keap)-1, from which it is released by electrophiles, reactive oxygen species (ROS), some natural compounds [1] and certain drugs, including tert-butylhydroquinone (tBHQ) [2,3,4] and bardoxolone methyl [5,6,7,8,9]. After phosphorylation, Nrf2 migrates to the nucleus, where it hybridizes with antioxidant response elements to transcribe a plethora of genes that protect tissues from reactive oxygen species (ROS) or damage [10]. Both the complex triterpenoid bardoxolone methyl [7] and the soluble, orally active small-molecular-weight tert-butylhydroquinone (tBHQ) [2] dissociate Nrf2 from Keap-1 and lead to its phosphorylation, stabilization and nuclear translocation to transcribe antioxidant genes. Indeed, we confirmed that the incubation of glomerular endothelial cells with tBHQ led to the transcription of many antioxidant genes [2], and that oral tBHQ given to a mouse model of oxidative stress, caused by an ongoing slow pressor infusion of Ang II, moderated the increase in mean arterial pressure (MAP) and the mesenteric arteriolar ROS and contractility at two weeks [3]. Both the cellular [2] and whole animal effects [3] of tBHQ were prevented by silencing or knocking out the Nrf2 gene. Thus, tBHQ provides a means to activate Nrf2 specifically and, thereby, to enhance the endogenous antioxidant defenses and moderate many of the adverse effects of Ang II. Surprisingly, we also observed that tBHQ led to an abrupt and significant increase in MAP over the first three days of Ang II infusion, which also depended on Nrf2 [3]. The present studies were designed to investigate the mechanism of these apparently paradoxical early effects of tBHQ using the same model as reported previously [3], but studied at day 3 of Ang II infusion and oral tBHQ administration.

Prolonged Ang II generates ROS that contribute to cardiovascular disease (CVD) [11], chronic kidney disease (CKD) [12] and hypertension [13,14]. Since effective antioxidants are not yet available for clinical use, the discovery of Nrf2 activators that can transcribe genes for endogenous antioxidant pathways has led to intense clinical interest [15]. Four different Nrf2 activators are being evaluated in >20 registered clinical trials [16]. There are reported benefits of Nrf2 activators in patients with neurological [17] and CV diseases [5] and diabetic nephropathy [8]. Unfortunately, the “Bardoxolone Methyl Evaluation in patients with Chronic Kidney Disease and Type 2 Diabetes Mellitus: the Occurrence of Renal Events (BEACON)” trial was terminated prematurely because of an increased number of adverse CVD events, including heart failure and increased BP. Remarkably, these became evident by the first follow-up visit at 2–4 weeks. Thereafter, only beneficial effects of increased glomerular filtration rate (GFR) were apparent [6]. Post hoc analyses of the BEACON trial have suggested that the adverse events were accompanied by increased endothelin 1 (ET1) signaling [18,19], but the absence of a faithful animal model has hindered the investigation of their mechanism and, thereby, the provision of rational means for their prevention.

We observed that the incubation of cultured vascular smooth muscle cells (VSMC) with tBHQ for 24 h causes a dose-dependent increase in the mRNA expression of cyclooxygenase 2 (COX2) (Figure 1). This extends reports that ROS in melanoma cells activate Nrf2, which enhances COX2 expression [20], that the exposure of zebrafish to tBHQ upregulates COX2 expression rapidly [4] and that Nrf2 activation enhances COX2 expression in the colons of mice fed a high-fat diet [21]. COX2 itself downregulates Nrf2 [22,23], indicating a negative feedback pathway. In contrast, Nrf2 activation in more severe and prolonged states of oxidative stress, as in mice administered lipopolysaccharide, reduces the expression of COX2 and inflammatory mediators [24]. However, the effects of Nrf2 activation on COX2 expression and function in the vasculature are not known and were, therefore, the goal of this study. 

COX2 metabolizes arachidonate to prostaglandin endoperoxide (PGH_2_), which can be metabolized further by thromboxane A_2_ synthase (TxA_2_S) to TxA_2_. Both PGH_2_ and TxA_2_ activate thromboxane prostanoid receptors (TPR) in blood vessels. We reported that TPR activation increases circulating endothelin 1 (ET-1) and enhances the ROS and contractility responses to ET1 in microvessels [25,26]. Moreover, Ang II induces ROS formation in VSMCs [27] that activate COX2 and TPR signaling in many models of CVD or CKD [25,26,28,29,30], while COX2 generation in mice increases Ang II-induced hypertension and causes oxidative stress, cardiac hypertrophy [31] and pressure-overload heart failure [32]. Nrf2 and COX2 are activated together in the vasculature in many models where COX2 enhances acute oxidative stress [33,34]. These models include the infusion of Ang II into normal human subjects [35] and experimental lead toxicity that enhances mesenteric arterial ROS and contractility though the activation of TPRs and angiotensin type 1 receptors [36]. Since an early activation of COX2 during Nrf2 therapy might explain the adverse effects observed in the BEACON trial, we tested the hypothesis that COX products signaling via TPRs mediate an increase in microarteriolar ROS and contractility during the early phase of Nrf2 activation. The results demonstrated that tBHQ increased mesenteric microarteriolar ROS and contractility to thromboxane and ET-1 accompanied by an increased expression of genes mediating ROS generation, and also thromboxane and ET-1 production or signaling. The functional effects of tBHQ could be ascribed to Nrf2 activation since they were absent in microarterioles from Nrf2 −/− mice, and to COX and TPR since they were absent in microarterioles from COX1 −/− mice given a COX2 blocker or in those given a TPR blocker.

## 2. Materials and Methods

***Animal Preparation***: Male C57Bl/6 mice weighing 25 to 30 g (Charles River Laboratory, Germantown, MD), Nrf2 −/− and +/+ mice (B6.129X1-Nfe2l2^tm1Ywk^/J, RRID:IMSR_JAX:017009, Jackson Lab, Bar Harbor, Hancock County, Maine, United States) [3] and COX1 −/− and +/+ mice [37] (all on a C57Bl/6 background, Jackson Laboratories, Bar Harbor, Maine) were maintained on tap water and standard chow (Na^+^ content 0.4 g·100 g^−1^; Harlan, Teklad). The protocol was terminated after 3 days, but otherwise followed closely the prior, more prolonged studies [3]. It conformed to the Guide for Care and Use of Laboratory Animals by the NIH Institute for Laboratory Animal Research and was approved by the Georgetown University Animal Care and Use Committee.

Mice were anesthetized with 2% isoflurane and prepared for a three-day subcutaneous infusion of Ang II at a slow pressor rate (400 ng·kg^−1^·min^−1^; Peninsula Laboratory, San Carlos, CA, USA) [38] via osmotic minipumps (model 1002; Alet, Palo Alto, CA, USA) [28] or sham procedures (sham). Mice infused with Ang II vs. sham were allocated randomly to receive oral tBHQ (0.1% in drinking water) vs. vehicle (0.06% ethanol). This dose of tBHQ activates microvascular Nrf2 signaling in mice and is well tolerated [3].

Mice were placed in a metabolic cage (days 2–3 of treatment) for a 24 h collection of urine for creatinine and 8-isoprostane F_2α_. Thereafter, they were sacrificed on day 3 for harvesting of mesenteric microarterioles for studies of ex vivo function. These were undertaken in 3 groups of mice (n = 6 for each): 

*Group 1*: used C57Bl/6 mice to test the effects of 3 days of tBHQ (vs. vehicle) on mesenteric microarteriolar contractions to phenylephrine (PE), thromboxane (U-46,619), endothelin 1 (ET1) and ET-1-induced ROS generation. We have reported previously that the BP of mice given tBHQ (vs. vehicle) and infused with Ang II (vs. sham) increased over 1–4 days, and that this increase was absent in Nrf2 −/− mice [3]. 

*Group 2*: used Nrf2 +/+ vs. −/− mice infused with Ang II to test the specificity of tBHQ as an activator of Nrf2 in its effects on the contractile and ROS responses in this protocol. 

*Group 3*: used mice infused with Ang II and given oral tBHQ to contrast responses in COX1 +/+ (vs. −/−) mice given parecoxib (10 mg·kg^−1^·d^−1^ × 3 d) in a maximally effective dose [39] (vs. vehicle) to test the roles of COX1 and 2 on mesenteric microarteriolar contractile and ROS responses. COX1 −/− mice were used to study responses to parecoxib to test COX2 independent of COX1. 

Microarterioles from an additional group of C57Bl/6 mice infused with Ang II and given oral tBHQ were incubated for 30 min with a fully effective concentration of SQ-29, 548 (10^−5^ mol·L^−1^) [30], to test the role of TPRs on the contractile and ROS responses of mesenteric microarterioles to ET1.

***Measurement of urinary 8-isoprostane F_2α_ and creatinine was*** as described [3].

***Preparation and study of mesenteric microarterioles:*** Vessels (luminal diameter 145 ± 10 μm, and length ~2 mm) were separated from the superior mesenteric bed, mounted on isometric wire myographs (M610, Danish Myotechnology A/S; Aarhus, Denmark) and studied as described [3]. 

***Mesenteric microarteriolar ET-1-induced ROS and contractions with phenylephrine (PE), thromboxane (U-46,619) and ET1:*** Arterioles were set to a wall tension of 0.2 mN·mm^−2^. ROS production was determined in vessels loaded with 5 × 10^−5^ mol·L^−1^ of dihydroethidium (DHE) from the change of fluorescence of ethidium (E) relative to DHE (E: DHE) with 10^−7^ mol·L^−1^ ET1 and quantitated by PTI RatioMaster^TM^ (Photon Technology International, London, ON, Canada), as described in [3]. Contractions to PE (10^−9^ to 10^−5^ mol·L^−1^), U-46,619 (10^−10^ to 10^−6^ mol·L^−1^) and ET-1 (10^−11^ to 10^−7^ mol·L^1^) were assessed relative to a standard contraction with norepinephrine (10^−5^ mol·L^−1^) plus KCl (123 mmol·L^−1^) (NAK), as described in [3]. 

Other vessels from mice receiving oral tBHQ and infused with Ang II for 3 days were incubated with SQ 29548 (10^−5^ mol·L^−1^, TPR inhibitor, Sigma, St. Louis, MO, USA) or vehicle for 30 min [26] to evaluate the role of TPRs in the increased contractions and ROS with ET 1 (10^−7^ mol·L^−1^).

***Chemicals and solutions:*** Reagents were from Millipore-Sigma Inc. (St. Louis, MO, USA). They are human Angiotensin II (Cat no: A9252), tBHQ (tert-butylhydroquinone, Cat no: 8.41424), SC-560 (Cat no: S2064), Paracoxib sodium (Cat no: SML2217), U-46,619 (Cat no: D8174-5mg), human endothelin-1 (Cat no: 05-23-3800), acetylcholine chloride (Cat no: A6625-10mg) and phenylephrine hydrochloride (Cat no: P6125-5G). Three fluorescence dyes were from Thermo Fisher, Waltham, MA, USA. They are DAF-FM Diacetate (Cat no: D23844), dihydroethdium (Cat no: DHE, D11347) and mitoSOX^tm^ red (Cat no: M36008). The physiological saline solution (PSS) was of the following composition (mM): NaCl 119, NaHCO_3_ 25, KCI 4.7, CaCl_2_ H_2_O 2.5, MgSO4-7H_2_O 1.17, KH_2_PO_4_ 1.18, Na_2_EDTA 0.026 and glucose 5.5. K-PSS was similar to PSS, except that NaCl was exchanged for KCI on an equimolar basis. PSS and KPSS were prepared by our lab, following the manufacturer’s instructions (Book: *Procedure for investigation of small vessels using small vessels myograph*. Page 46, Danish Myo Technology Inc., Aarhus, Denmark).

***Statistical analysis:*** The number of mice used for each protocol (n = 6 per group) was selected after a power analysis based on our prior data with this model [3]. Data are presented as mean ± SEM. Cumulative concentration-response experiments were analyzed by nonlinear regression (curve fit), and differences were assessed by a two-way, repeated-measures analysis of variance (ANOVA) to assess the effects of ANG II vs. tBHQ (group 1) or Nrf2 +/+ vs. −/− (group 2) or COX1 +/+ vs. −/− or COX1 −/− with vehicle vs. parecoxib (group 3). A probability value <0.05 was considered statistically significant. Gene and protein expression studies were conducted in triplicate, and differences between experimental groups were compared by ANOVA.

## 3. Results

The effects of incubating vascular smooth muscle cells (VSMCs) with tBHQ (vs. vehicle) on the mRNA expression of COX2 (Figure 1) was studied by incubation of VSMCs [27] for 24 h that with tBHQ was found to lead to dose-dependent increases in the mRNA expression of COX2. 

The effects of oral tBHQ (vs. vehicle) and Ang II infusion (vs. sham) for three days on the mRNA expression of COX1 and 2, p47^phox^, endothelin type A and B receptors (ETAR and ETBR), thromboxane A_2_ synthase (TxA_2_S) and TPR was studied in mouse mesenteric microarterioles (Table 1). Infusions of Ang II increased the expression of COX1 and 2, p47^phox^, ETAR, TxA_2_S and TPR, while tBHQ enhanced the expression of COX1 and 2, p47^phox^ and TPR. Moreover, tBHQ enhanced the effect of Ang II to increase p47^phox^, ETAR and TxA_2_S. We conclude that tBHQ potentiates the effects of Ang II to increase the microarteriolar expression of the genes whose products can generate ROS or mediate signaling via ETAR and TPR. 

The effects of knockout of the Nrf2 and COX1 genes and the blockade of COX2 on the excretion of 8-isoprostane F_2α_ relative to creatinine during three days of oral tBHQ and Ang II infusion (Table 2) demonstrate that the excretion of 8-isoprostane F_2α_ relative to creatinine was increased by the infusion of tBHQ and by Ang II, while tBHQ enhanced the effects of Ang II infusion to increase 8-isoprostane F_2α_ excretion (Table 2A). These effects of tBHQ to enhance 8-isoprostane F_2α_ excretion during the infusion of Ang II were lost in Nrf2 −/− mice (Table 2B), were reduced in COX1 −/− (vs. +/+) mice and reduced further in mice treated with parecoxib to block COX2 (Table 2C). We conclude that the activation of Nrf2 by tBHQ potentiates the effects of Ang II infusion to increase oxidative stress, and that this depends on COX1 and COX2.

The effects of oral tBHQ (vs. vehicle) and Ang II infusion (vs. sham) for three days on mesenteric microarteriolar contractions and with PE, U-46,619 and ET1 and ROS generation with ET-1 (Figure 2; Table 3A–C) include that oral tBHQ administration or infusion of Ang II for 3 days in mice of group 1 did not change the microarteriolar contractions to phenylephrine (Table 3A). Therefore, other protocols did not include PE. Although tBHQ administration or Ang II infusion alone did not change contractions to U-46,619 or ET1 or ROS generation with ET1, all of these were enhanced by tBHQ given during an Ang II infusion (Figure 2; Table 3A). The effects of tBHQ to enhance contractions to thromboxane and ET-1 and ROS generation with ET-1 during Ang II infusion were lost in microarterioles from Nrf2 −/− mice of group 2 (Figure 3; Table 3B), and were reduced in microarterioles from COX1 −/− (vs. +/+) mice of group 3 (Figure 4; Table 3C) and reduced further in both COX1 +/+ and −/− arterioles from mice given parecoxib to block COX2 (Figure 4; Table 3C). We conclude that neither 3 days of oral tBHQ administration nor Ang II infusion alone modify contractions to PE, thromboxane or ET1 or ROS generation with ET1, but that contractions to thromboxane and ET-1 and ROS generation are enhanced by tBHQ given during an Ang II infusion by a mechanism that depends on Nrf2, COX1 and 2. 

The effects of the TPR blockade on ROS generation and contractions to ET-1 in mice given tBHQ and infused for three days with Ang II (Figure 5) demonstrate that, as before, neither tBHQ administration alone nor Ang II infusion alone changed the ROS generation or contractions with ET1, but these were both increased by tBHQ given during Ang II infusion. These increases in response to tBHQ during Ang II were prevented in vessels incubated with a maximally effective concentration of SQ-29,548 (10^−5^ mol·L^−1^; Figure 5) for 30 min to block TPRs [25]. We conclude that the enhanced ROS generation and contractions with ET1 in response to tBHQ administration during 3 days of Ang II infusion depend on TPRs.

## 4. Discussion

We reported that the MAP of conscious, unrestrained mice was increased by tBHQ during the first 3 days of an infusion of Ang II, yet was reduced during the last days (11–13) and that both the early increase and the later reduction in MAP with tBHQ were lost in Nrf2 −/− mice [3]. The new findings relate to the early, 3-day period of tBHQ administration and infusion of Ang II while the MAP was increased. At this time, tBHQ administration enhanced the effect of Ang II infusion to increase the microarteriolar mRNA expression for p47^phox^, ETAR and TxA_2_S, and the generation of ROS assessed both by excretion of 8-isoprostane F_2α_ and by ethidium:dihydroethidium fluorescence of microarterioles incubated with ET1. However, these effects of tBHQ were lost in microarterioles from Nrf2 −/− (vs. +/+) mice, were blunted in COX1 −/− (vs. +/+) and blunted further in mice given a COX2 blocker. These effects of tBHQ to enhance ROS were accompanied by enhanced contractions to U-46,619 and ET1 that also were lost in Nrf2 −/− mice and were reduced in COX1 −/− mice and mice given parecoxib to block COX2. Finally, the effects of tBHQ administration to enhance ET1 contractions and ROS in microarterioles from mice infused with Ang II for 3 days were prevented by the blockade of TPRs. Thus, the early 3-day increase in MAP with tBHQ during the infusion of Ang II activates Nrf2, which enhances the microarteriolar mRNA expression of p47^phox^ accompanied by an enhanced generation of ROS, with enhanced microarteriolar mRNA expression of COX1 and 2, TxA_2_S and TPR accompanied by enhanced contractions with thromboxane and with enhanced microarteriolar mRNA expression of ETARs accompanied by enhanced contractions and ROS generation with ET1. These studies implicate both COX1 and COX2 since tBHQ during Ang II upregulated mRNA for both COX isoforms and the enhanced microarteriolar contractions to U-46,619 and ET-1 and ROS generation with ET-1 were all reduced in COX1 −/− (vs. +/+) mice and mice given parecoxib. There was a specific role for COX2 since any off-target effects of parecoxib on COX1 would be absent in microarterioles from COX1 −/− mice. Finally, we found dose-dependent upregulation of COX2 by tBHQ in cultured VSMCs. These early effects of tBHQ during Ang II are summarized in Figure 6. They contrast strikingly with the opposite, beneficial effects observed after 11–13 days of tBHQ administration during the Ang II infusion using a similar, but more prolonged, model [3]. 

The novel finding that tBHQ enhances the expression of the mRNA for COX2 in cultured VSMCs and in microarterioles of mice infused with Ang II for three days is consistent with the reports that the gene for COX2 contains a consensus antioxidant response element at the Nrf2 binding site [40], and with reports that exposure to tBHQ upregulates COX2 in zebrafish [4] and in tumor cells [20,40]. Moreover, the finding that it also enhances ROS in the systemic circulation (indexed by enhanced excretion of 8-isoprostane F_2α_) and the vasculature (indexed by ethidium:dihydroethidium fluorescence of mesenteric microarterioles) that are dependent on COX1+2 and TPR is consistent with our prior finding that the COX/TPR pathway acts in a positive feedback mode [30] to enhance the effects of Ang II to increase microvascular ROS and contractility in rodents [26], and with other studies that have reported specific roles for COXs, Nrf2 and tBHQ in the responses to oxidative stress or in the production of ROS in vascular models [36,41,42,43].

There were no increases in BP or microvascular ROS or contractility at the early, three-day phase of infusion of Ang II, in contrast to the marked increases observed in previous studies of Ang II-infused rodents at 2 weeks [3,26,28,30,38,44]. The administration of tBHQ alone over 3 days in this study or over 11–13 days in a previous study [3] did not change BP, ROS or contractility. However, these did increase when tBHQ was given during a 3-day infusion of Ang II alone but decreased after 11–13 days. These time-dependent responses to tBHQ are consistent with an early increase in systemic and microvascular ROS at 3 days, but a reduction at 14 days [3] since oxidative stress enhances the rise in BP and microvascular contractions to thromboxane and ET-1 with Ang II [13,14,28,30,38,44]. The sources of the early microvascular oxidative stress with tBHQ during Ang II infusion likely included increases in NADPH oxidase, since the expression of the gene for its p47^phox^ component was increased, and increases in ROS production by mitochondria, as indexed by increased MitoSOX^TM^ red fluorescence microarterioles. An increase in microvascular NADPH oxidase and mitochondrial ROS generation, in turn, may be a response to increased TPR activation since this enhances microvascular NADPH oxidase expression [44] and uncouples endothelial nitric oxide synthase to generate superoxide [45]. Moreover, TPR activation increases ET-1 generation in concert with microvascular ETAR signaling [25], which can reinforce the generation of oxidative stress [46]. 

There is a more complex interaction between COX products and Ang II [47]. Thus, COX2 can be induced by Ang II in VSMCs [48] and can enhance Ang II-induced [47] and renovascular hypertension [29] in some studies; yet, COX products can moderate hypertension in others [49] and a COX blockade with non-steroidal anti-inflammatory agents usually increases the BP of patients with hypertension. These discordant effects of COX products may depend on the balance between the production of vasoconstrictor, pro-hypertensive vs vasodilator, anti-hypertensive prostaglandins [50]. Thus, the early upregulation of the microvascular genes for TxA_2_S and TPR by tBHQ after 3 days of Ang II infusion in this study should be contrasted with the reduction in the excretion of TxB_2_ and 8-isoprostane F_2α_ by oral tBHQ after 11–13 days of Ang II infusion in our previous study [3]. This raises the possibility that time-dependent changes in COX activity and TxA_2_ production may underlie the time-dependent effects of Nrf2 activation during Ang II infusion. Interestingly, the promoter for the gene for TxA_2_S contains an activating Nrf2 binding site [51] that could mediate the early expression of TxA_2_S in this study. Heart failure activates tissue M1 macrophages [52] that are major sites of expression of TxA_2_S [53], thereby providing a potential explanation for the propensity of patients with heart failure to experience adverse CVD events during the initiation of bardoxolone methyl therapy [54]. However, more prolonged Nrf2 activation promotes macrophage transition to the M2 phenotype via the activation of peroxisome proliferator-activated receptor gamma (PPAR***γ***) [55], which itself suppresses TxA_2_S gene expression via a redox-independent mechanism [56]. This could explain the reduced TxB_2_ excretion in mice in this model, studied after 11–13 days of Nrf2 activation [3]. Further studies will be required to test whether an early increase in TPR signaling after Nrf2 activation during Ang II, which promotes vasoconstriction, does indeed switch to a predominant signaling by other prostaglandins such as prostacyclin [57], PGE2 [58] and PGD2 [59], which can moderate microvascular contractility in the longer term. 

We acknowledge some limitations to our study. First, we used tBHQ rather than bardoxolone methyl, which is the drug selected to activate Nrf2 in most clinical trials. However, tBHQ is non-toxic, since it did not modify the gain in body weight of the mice [3] and is added to human food as a preservative [60]. Its actions can be attributed to the activation of Nrf2 since it translocates Nrf2 to the nucleus and initiates Nrf2 signaling [2], and its effects are prevented by the deletion of Nrf2 [2,3]. In contrast, bardoxolone methyl undergoes complex chemical transformations [7], has variable effects in rodents and contaminants have obscured its action in prior functional studies [37]. Therefore, tBHQ was selected for our studies. Second, we measured mRNA rather than protein expression. However, the corresponding functional proteins were apparently expressed since an increased microarteriolar mRNA expression of p47^phox^ was accompanied by an increased microarteriolar ROS, and an increased microarteriolar mRNA for TPR and ETAR were accompanied by an increased microarteriolar contractility to thromboxane and ET1. These were specific effects, since the contractile responses to PE were unchanged. Moreover, increases in the microarteriolar expression of genes that enhance signaling via ROS, thromboxane and ET-1 are consistent with reports that these mediate increases in the BP of mice infused with Ang II in other studies [28,31,44,61]. 

## 5. Conclusions

In conclusion, the increase in MAP of mice given tBHQ over the first three days of a slow pressor infusion of Ang II is associated with enhanced microvascular ROS and contractility to thromboxane and ET1. These changes depend on Nrf2, COX1 and 2 products and TPR signaling (Figure 6). Although further work is required to identify the molecular mechanisms that underlie the time-dependent switch in microvascular responses to Nrf2 activators in this model, other studies have reported that the activation of Nrf2 during mild oxidative stress, as during the early phase of the slow pressor infusion of Ang II, enhances the expression of COX2 [4,20,21], whereas activation of Nrf2 during more severe or prolonged oxidative stress, as after 2 weeks of Ang II infusion [3], reduces the expression of COX2 [24]. Whether such time-dependent changes in the microarteriolar expression of COX2 products and TxA_2_ generation in this model of a slow, sustained infusion of Ang II may switch the early pro-oxidant effects of Nrf2 to later antioxidant responses requires further study. 

### Perspective

Patients with diabetic nephropathy given bardoxolone methyl to activate Nrf2 in the BEACON trial had a sustained 25% increase in their GFR, yet the trial was stopped prematurely because of adverse CVD events, including hypertension and heart failure, which were noted within the first follow-up period of 2–4 weeks [6]. Animal models have demonstrated an increased activity of angiotensin II in diabetic nephropathy [61,62], which was modelled in our study by a slow pressor infusion of Ang II. The results of this study suggest that an undetected early increase in COX-dependent TPR signaling might have contributed to the adverse events in patients with diabetic nephropathy in the BEACON trial. Thus, studies of the responses to an infusion of U-46,619 to activate TPRs in experimental animals report that TPR activation enhances Na^+^ reabsorption in the loop of Henle [63], reduces Na^+^ excretion and causes salt-sensitive hypertension [64], increases renal [65], peripheral and pulmonary vascular resistances [66], reduces cardiac output [67], enhances coronary artery endothelial dysfunction and contractility [68] and induces platelet aggregation [69]. Thus, the activation of TPRs could underlie an increase in vascular resistance, Na^+^ retention and BP with coronary artery vasoconstriction and fluid retention that could precipitate heart failure in susceptible patients. New clinical trials of bardoxolone methyl and other drugs that activate Nrf2 have been launched [8] (for example, N CT 03019185 and N CT 03366337), yet clear knowledge of the underlying mechanisms that can mediate the early adverse CVD events remain uncertain. This mouse model generally recapitulates the time course of clinical events during Nrf2 activation with bardoxolone methyl in patients with diabetic nephropathy [6], where an early rise in BP with adverse cardiovascular events are succeeded by a beneficial response [3], and provides a new model to study the underlying mechanisms.

## Figures and Tables

**Figure 1 antioxidants-11-00845-f001:**
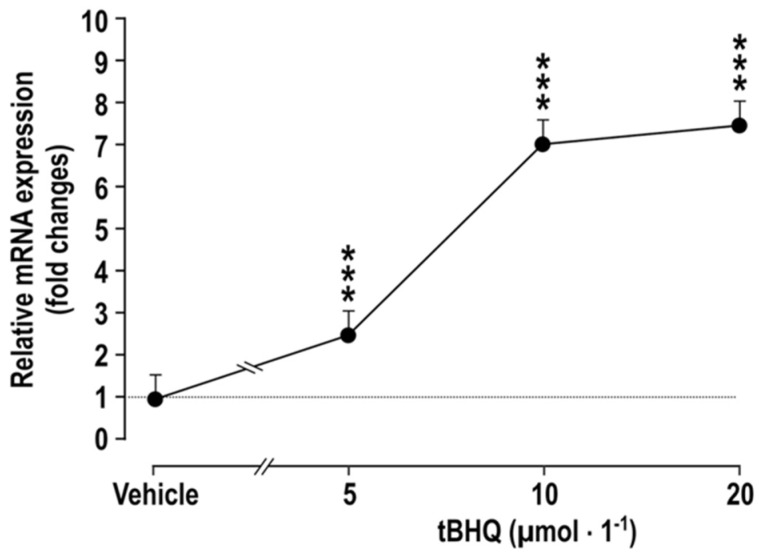
Incubation of vascular smooth muscle cells with tBHQ for 24 h causes dose-dependent increases the mRNA expression of COX2, compared to vehicle-treated controls. Mean ± SEM values (*n* = 5). Compared to vehicle: ***, *p* < 0.005.

**Figure 2 antioxidants-11-00845-f002:**
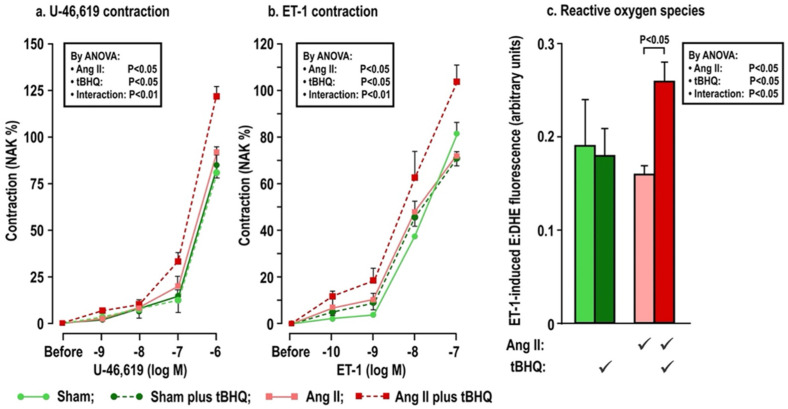
Oral tBHQ given to mice during a three day infusion of angiotensin II enhances the contractions to thromboxane (Panel **a**) and endothelin-1 (Panel **b**) and the reactive oxygen species generation with endothelin-1 (Panel **c**) in mesenteric microarterioles from C57Bl/6 mice. Mean ± SEM values (n = 6 per group) in mesenteric microarterioles from mice of group 1 depicting dose-dependent contraction to thromboxane and endothelin 1 and reactive oxygen species generation from the ratio of ethidium to dihydroethidium fluorescence during incubation with endothelin 1 (10^−7^ mol·L^−1^). ET-1, endothelin 1; ROS, reactive oxygen species; NAK, standard contraction with norepinephrine + KCl; tBHQ, tert-butylhydroquinone (0.1% added to drinking water × 3 days); Ang II (angiotensin II infusion at 400 ng·Kg^−1^·min^−1^ sc × 3 days).

**Figure 3 antioxidants-11-00845-f003:**
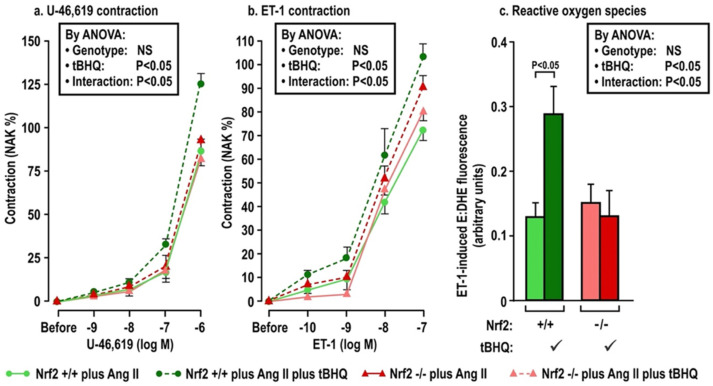
Oral tBHQ given to mice during a three-day infusion of angiotensin II enhances the contractions to thromboxane (Panel **a**) and endothelin I (Panel **b**) and the reactive oxygen species generation with endothelin 1 (Panel **c**) in mesenteric microartioles from Nrf2 +/+ mice, but fails to enhance these in arterioles from Nrf2 −/− mice. Mean ± SEM values (n = 6 per group) in mesenteric microarterioles from Nrf2 wild type (+/+) or knockout (−/−) mice of group 2 given tBHQ (0.1% × 3 days) vs. vehicle. These mice were all infused with angiotensin II (400 ng·kg^−1^·d^−1^ SC × 3 days). Nrf2, nuclear factor erythroid factor E2-related factor 2. See legend to Figure 2.

**Figure 4 antioxidants-11-00845-f004:**
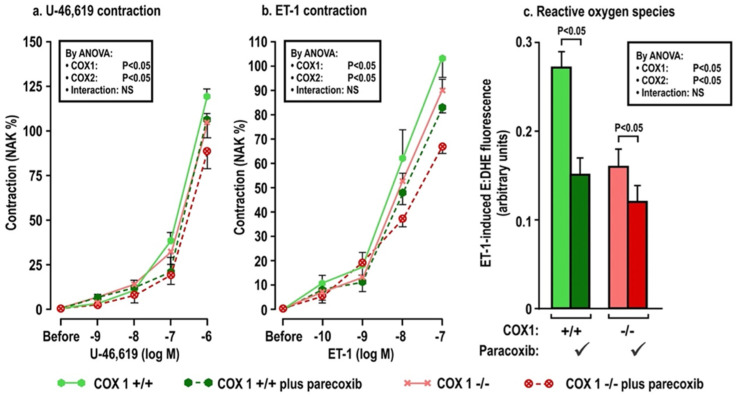
Oral tBHQ given to mice during a three-day infusion of angiotensin II enhances the contractions to thromboxane (Panel **a**), endothelin 1 (Panel **b**) and the reactive oxygen species generation with endothelin I (Panel **c**) in mesenteric microarterioles from COX1 +/+ mice, but these responses are diminished in microarterioles from COX1 −/− mice and in mice given parecoxib for 3 days to block COX2. Mean ± SEM values (n = 6 per group) in mesenteric microarterioles of cyclooxygenase 1 wild type (+/+) or knockout (−/−) mice given parecoxib (10 mg·kg·d^−1^ × 3 days to block COX2) vs. vehicle in mice of group 3. These mice all received oral tert-butylhydroquinone (0.1% × 3 days) and were infused with angiotensin II (400 ng·kg^−1^·d^−1^ SC × 3 days). COX, cyclooxygenase. See legend to Figure 2.

**Figure 5 antioxidants-11-00845-f005:**
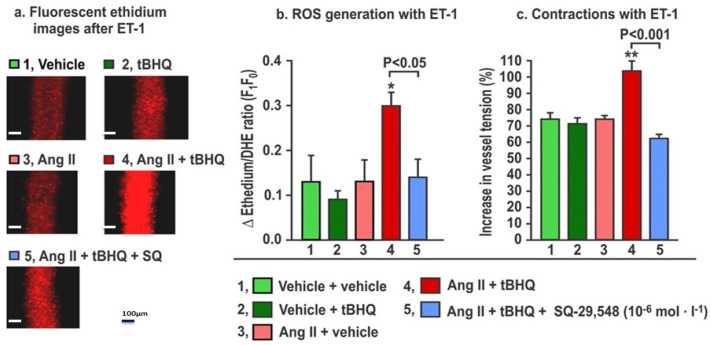
The increases in reactive oxygen species (ROS) generation and contractions with endothelin-1 in mesenteric microarterioles from mice given tBHQ (0.1% × 3 days) vs. vehicle and infused with angiotensin II (400 ng·kg^−1^·d^−1^ SC × 3 days) vs. sham have increased reactive oxygen species (Panels **a** and **b**) and contractions to endothelin 1 (Panel **c**) that are prevented by incubation for 30 min with SQ-29,584 (10^−5^ mol·L^−1^) to block thromboxane prostanoid receptors. Fluorescent microscopy images of ethidium fluorescence (Panel **a**) and mean ± SEM values for ROS generation (Panel **b**) and contractions (Panel **c**) with endothelin 1 (ET1, 10^−7^ mol·L^−1^) in mesenteric microarterioles from mice (n = 6).

**Figure 6 antioxidants-11-00845-f006:**
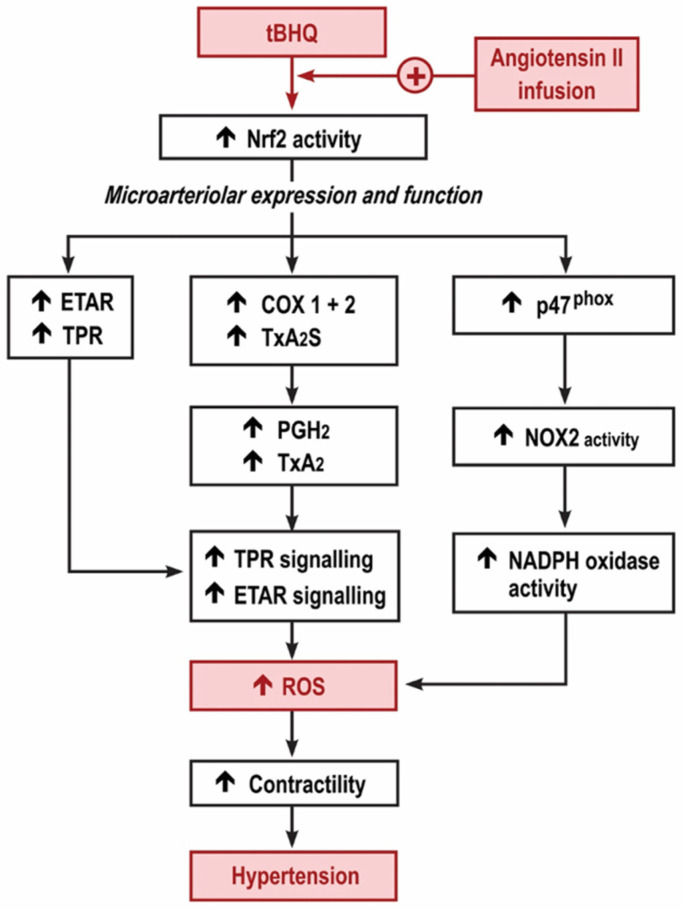
Graphical depiction of the hypothesis for the early increase in reactive oxygen species and microvascular dysfunction with tert-butylhydroquinone during the angiotensin II infusion. Nrf2, nuclear factor erythroid factor E2-related factor 2; TPR, thromboxane prostanoid receptors; ETAR, endothelin type A receptors; COX, cyclooxygenase; TxA_2_S, thromboxane A_2_ synthase; NOX, neutrophil oxidase; NADPH, nicotinamide adenine dinucleotide phosphate.

**Table 1 antioxidants-11-00845-t001:** mRNA expression in mouse mesenteric microarterioles; effects of 3 days of angiotensin II (Ang II) infusion or tBHQ administration.

Gene	Vehicle		tBHQ		By ANOVA, Effect of		
	Sham	Ang II	Sham	Ang II	Ang II	tBHQ	Interaction
**COX1**	7.88 ± 4.82	59.87 ± 18.00 **	58.34 ± 9.51 ‡‡	145.13 ± 23.15 *‡	*p* < 0.01	*p* < 0. 01	NS
**COX2**	0.97 ± 0.10	2.73 ± 0.68 *	2.19 ± 0.47 ‡‡	6.04 ± 0.90 *‡	*p* < 0.01	*p* < 0.01	NS
**p47^phox^**	1.00 ± 0.27	1.33 ± 0.15 *	1.69 ± 0.85 ‡	3.21 ± 0.57 *‡	*p* < 0.05	*p* < 0.05	*p* < 0.05
**ETAR**	1.15 ± 0.10	1.69 ± 0.18 *	1.08 ± 0.29	3.13 ± 0.45 **‡	*p* < 0.01	*p* < 0.05	*p* < 0.05
**ETBR**	1.15 ± 0.08	1.48 ± 0.63	1.03 ± 0.14	0.40 ± 0.11‡	NS	NS	NS
**TxA_2_S**	0.82 ± 0.11	1.51 ± 0.40 *	1.06 ± 0.39	3.63 ± 0.55 *‡	*p* < 0.01	*p* < 0.05	*p* < 0.05
**TPR**	0.90 ± 0.12	3.45 ± 0.72 **	2.43 ± 0.75 ‡	6.23 ± 0.33 **‡	*p* < 0.01	*p* < 0.01	NS

Mean ± SEM values (n = 5 per group) for mRNA expression for cyclooxygenase 1 and 2; protein 47 phagocyte oxidase; endothelin type A and B receptors; thromboxane A_2_ synthase; thromboxane-prostanoid receptor. Compared to sham: *, *p* < 0.05; **, *p* < 0.01; Compared to vehicle: ‡, *p* < 0.05; ‡‡, *p* < 0.01.

**Table 2 antioxidants-11-00845-t002:** Renal excretion of 8-isoprostane F_2α_ relative to creatinine in conscious mice.

**A.** **Effects of tBHQ and Ang II infusion in group 1**						
**Vehicle**		**tBHQ**		**By ANOVA, effect of**		
**Sham**	**Ang II**	**Sham**	**Ang II**	**Ang II**	**tBHQ**	**Interaction**
0.88 ± 0.10	1.10 ± 0.10	1.24 ± 0.15 ‡	2.56 ± 0.18 *‡	*p* < 0.05	*p* < 0.05	*p* < 0.05
**B.** **Effects of Nrf2 gene on tBHQ responses in Ang II infused mice in group 2**						
**Ang II + Vehicle**		**Ang II + tBHQ**		**By ANOVA, effect of**		
**Nrf2 +/+**	**Nrf2 −/−**	**Nrf2 +/+**	**Nrf2 −/−**	**Nrf2**	**tBHQ**	**Interaction**
0.95 ± 0.15	1.05 ± 0.11	1.96 ± 0.12 ‡	1.14 ± 0.25 *	*p* < 0.05	*p* < 0.05	*p* < 0.05
**C.** **Effects of COX2 blockade in COX1 +/+ or −/− mice given tBHQ and infused with Ang II in group 3**						
**Ang II + tBHQ + Vehicle**		**Ang II + tBHQ + parecoxib**		**By ANOVA, effects of**		
**COX1 +/+**	**COX1 −/−**	**COX1 +/+**	**COX1 −/−**	**COX1**	**Parecoxib**	**Interaction**
2.58 ± 0.15	1.70 ± 0.12 *	1.64 ± 0.14 ‡	1.26 ± 0.15 *‡	*p* < 0.05	*p* < 0.05	*p* < 0.05

Mean ± SEM values (n = 5 per group) for mRNA expression for cyclooxygenase 1 and 2; protein 47 phagocyte oxidase; endothelin type A and B receptors; thromboxane A_2_ synthase; thromboxane-prostanoid receptor. Compared to sham: *, *p* < 0.05; Compared to vehicle: ‡, *p* < 0.05.

**Table 3 antioxidants-11-00845-t003:** Maximum contractions to phenylephrine, thromboxane (U-46,619) and endothelin 1 and ROS generation with endothelin 1 in mesenteric microarterioles: effects of tert-butylhydroquinone (tBHQ) administration and angiotensin II infusion (Ang II; Panel A) for three days, of tBHQ administration to Nrf2 +/+ (vs. −/−) mice infused with Ang II for three days (Panel B) and of cyclooxygenase 1 −/− (vs. +/+) with blockade of cyclooxygenase 2 with parecoxib (vs. vehicle) in mice given tBHQ and infused with angiotensin II for 3 days (Panel C).

**A.** **Effects of tBHQ and Ang II in group 1**							
**Responses**	**Vehicle**		**tBHQ**		**By ANOVA, effect of**		
	**Sham**	**Ang II**	**Sham**	**Ang II**	**Ang II**	**tBHQ**	**Interaction**
Phenylephrine contraction (%)	67.5 ± 5.5	72.3 ± 5.9	74.4 ± 4.4	73.5 ± 6.3	NS	NS	NS
U-46,619 contraction (%)	85.1 ± 5.9	91.9 ± 2.7	81.4 ± 3.6	120.0 ± 5.5 *‡	*p* < 0.05	*p* < 0.05	*p* < 0.05
ET-1 contraction (%)	80.9 ± 4.5	71.4 ± 2.3	71.3 ± 3.3	102.7 ± 7.9 *‡	*p* < 0.05	*p* < 0.05	*p* < 0.05
ET1-induced ROS (Δ unit)	0.19 ± 0.05	0.18 ± 0.03	0.16 ± 0.02	0.27 ± 0.02 *‡	*p* < 0.05	*p* < 0.05	*p* < 0.05
**B.** **Effects of Nrf2 genotype on responses to tBHQ in mice infused with Ang II for 3 days in group 2**							
**Responses**	**Ang II + Vehicle**		**Ang II + tBHQ**		**By ANOVA, effect of**		
	**Nrf2 +/+**	**Nrf2 −/−**	**Nrf2 +/+**	**Nrf2 −/−**	**Genotype**	**tBHQ**	**Interaction**
U-46,619 contraction (%)	87.1 ± 25.9	82.5 ± 3.7	124.9 ± 6.5 ‡	93.9 ± 2.8	NS	*p* < 0.05	*p* < 0.05
ET-1 contraction (%)	80.3 ± 4.5	85.9 ± 3.6	102.7 ± 5.8 ‡	90.4 ± 5.3	NS	*p* < 0.05	*p* < 0.05
ET-1-ROS (Δ unit)	0.13 ± 0.04	0.15 ± 0.03	0.29 ± 0.05 ‡	0.13 ± 0.06	NS	*p* < 0.05	*p* < 0.05
**C.** **Effects of cyclooxygenase 1 genotype and cyclooxygenase 2 blockade in mice given tBHQ and infused with angiotensin II for 3 days**							
**Responses**	**tBHQ + Ang II + Vehicle**		**tBHQ + Ang II + Parecoxib**		**By ANOVA, effect of**		
	**COX1 +/+**	**COX1 −/−**	**COX1 +/+**	**COX1 −/−**	**COX1**	**Parecoxib**	**Interaction**
U-46,619 contraction (%)	120.0 ± 3.7	105.5 ± 5.0 *	102.5 ± 4.9 ‡	87.9 ± 8.5 *‡	*p* < 0.05	*p* < 0.05	NS
ET-1 contraction (%)	102.7 ± 7.8	90.3 ± 4.38 *	83.4 ± 2.3 ‡	67.3 ± 3.3 *‡	*p* < 0.05	*p* < 0.05	NS
ET-1-ROS (Δ unit)	0.26 ± 0.03	0.16 ± 0.3 *	0.15 ± 0.02 ‡	0.12 ± 0.01 ‡	*p* < 0.05	*p* < 0.05	NS

Mean ± SEM values (n = 6 per group); ET-1, endothelin 1; ROS, reactive oxygen species. Comparing columns 2 with 1 and 4 with 3: *, *p* < 0.05. Comparing columns 3 with 1 and 4 with 2: ‡, *p* < 0.05.

## Data Availability

Data is contained within the article.

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
