# Peer review of "Activation of Nrf2 in Mice Causes Early Microvascular Cyclooxygenase-Dependent Oxidative Stress and Enhanced Contractility"

_antioxidants, 2022, doi:10.3390/antiox11050845_

Round 1
Reviewer 1 Report
Please see attached file.

Author Response
We thank you for your appreciative comments. There were no specific questions to answer.
Reviewer 2 Report
Review of manuscript by Wang et al., titled “Activation of Nrf2 in Mice Causes Early Microvascular Cyclooxygenase-dependent Oxidative Stress and Enhanced Contractility”.
Authors have put efforts to rewrite the manuscript with more details. There are few minor concerns which are not addressed
- Please provide the exact mouse strain and stock number of mouse models from the Jax laboratory in the methods section.
- Chemicals and solutions: provide the catalog number and name of important chemical reagents used in the study.
- Scale bar for the images missing.
Author Response
We thank you also for your appreciation.
Minor comments:
- “Please provide the exact mouse strain and stock number of mouse models from the Jax laboratory in the methods section.” The revised manuscript states: Male C57Bl/6 mice weighing 25 to 30 g (Charles River Laboratory, Germantown, MD), Nrf2 -/- and +/+ mice (B6.129X1-Nfe2l2tm1Ywk/J, RRID:IMSR_JAX:017009, Jackson Lab, USA).
- “Chemicals and solutions: provide the catalog number and name of important chemical reagents used in the study.” We now say: Chemicals and solutions: Reagents were from Millipore-Sigma Inc (St. Louis, MI). They are human Angiotensin II (Cat no:A9252), tBHQ (tert-butylhydroquinone, Cat no: 8.41424 ), SC-560(Cat no: S2064), Paracoxib sodium (Cat no:SML2217)), U-46,619 (Cat no:D8174-5mg), human endothelin-1 (Cat no:05-23-3800), acetylcholine chloride (Cat no: A6625-10mg), phenylephrine hydrochloride (Cat no:P6125-5G). Three fluorescence dye were from Thermo Fisher, USA, They are DAF-FM Diacetate (Cat no: D23844 ). dihydroethdium (Cat no: DHE, D11347)), mitoSOXtm red( Cat no: M36008). The physiological saline solution (PSS) was of the following composition (mM): NaCl 119, NaH2CO3 25, KCI 4.7, CaCl2 H2O 2.5, MgSO4-7H2O 1.17, KH2PO4 1.18, Na2EDTA 0.026 and glucose 5.5. K-PSS was similar to PSS except that NaCl was exchanged for KCI on an equimolar basis. PSS and KPSS were prepared by our lab following the manufacturer's instruction (Book: Procedure for investigationn of small vessels using small vessels myograph. Page 46, Danish Myo technology Inc. Denmark).
- The scale bar has been added to fig 5a.
This manuscript is a resubmission of an earlier submission. The following is a list of the peer review reports and author responses from that submission.
Round 1
Reviewer 1 Report
Please see attached file.

Reviewer 2 Report
In the submitted manuscript the authors attempt to explain the molecular mechanism underlying an apparent paradox. In fact, Nrf2 is a transcription factor that induces the activation of fundamental genes in antioxidant defense. The tBHQ induces the activation of this protein, but the authors report how the simultaneous infusion of tBHQ and Ang II leads to an increase in blood pressure in the first 3 days.
General comment:
The manuscript turns out to be very confusing. Since the topic is complex, the authors should explain it more clearly. In particular:
- Amplify the chemicals and solutions paragraph.
- Improve the comments of the results. In addition to better compare the early and late effects of the tested compound, a time point longer than 3 days should be explored.
- Appropriate information on some molecules, indicated in the text simply as abbreviations, should be provided.
Reviewer 3 Report
Review of manuscript by Wang et al., titled “Activation of Nrf2 in Mice Causes Early Microvascular Cyclooxygenase-Dependent Oxidative Stress and Enhanced Contractility”.
The authors are investigated a very important question regarding the failure of protection from Nrf2 activation in the BEACON trial. And early effects of Nrf2 activation.
Authors have done a lot of experiments. However, the results are not present well and hard to follow.
The major concern is:
- Results are too short. Authors should elaborate the result section. One liner result description won’t help readers.
- All the groups should be mentioned clearly in the result sections. Increased or decreased expression in compared to which group etc.
- Labeling of figures and graphs are confusing. Its not easy to follow. Please revise all the figures with proper easy to follow labelling.
- Discussion section is not helpful at all. It has to be revised extensively. Authors need to discuss their finding in details.
Minor comments:
- Line 98; Animal preparation. Please change or just keep “Animals”
- Please provide the catalog/Stock number of mouse models in the methods section.
- Scale bar for the images missing.
- Figure description on the top of figures are not required. Put it in the legend or result description.